# Impact of Appropriate Empirical Antibiotic Treatment on the Clinical Response of Septic Patients in Intensive Care Unit: A Single-Center Observational Study

**DOI:** 10.3390/antibiotics13060569

**Published:** 2024-06-19

**Authors:** Mateo Tićac, Tanja Grubić Kezele, Marina Bubonja Šonje

**Affiliations:** 1Department of Anesthesiology and Intensive Care, Clinical Hospital Center Rijeka, 51000 Rijeka, Croatia; mateo.ticac@uniri.hr; 2Department of Anesthesiology, Reanimatology, Intensive Care and Emergency Medicine, Faculty of Medicine, University of Rijeka, 51000 Rijeka, Croatia; 3Department of Clinical Microbiology, Clinical Hospital Center Rijeka, 51000 Rijeka, Croatia; marina.bubonja@uniri.hr; 4Department of Physiology, Immunology and Pathophysiology, Faculty of Medicine, University of Rijeka, 51000 Rijeka, Croatia; 5Department of Microbiology and Parasitology, Faculty of Medicine, University of Rijeka, 51000 Rijeka, Croatia

**Keywords:** antibiotics, clinical outcome, empirical therapy, intensive care unit, mortality, sepsis

## Abstract

The appropriate antibiotic treatment of patients with bacterial sepsis in the intensive care unit (ICU) remains a challenge. Considering that current international guidelines recommend 7 days of antibiotic therapy as sufficient for most severe infections, our primary outcome was a comparison of clinical response to initial empirical therapy on day 7 and mortality between two groups of septic patients—with appropriate (AEAT) and inappropriate (IEAT) empirical antibiotic therapy according to the in vitro sensitivity of bacteria detected in a blood culture (BC). Adult patients admitted to the ICU between 2020 and 2023, who were diagnosed with sepsis according to the Sequential Organ Failure Assessment (SOFA) score ≥ 2 in association with a suspected or documented infection, were selected for the study. Of the 418 patients, 149 (35.6%) died within 7 days. Although the AEAT group had a lower mortality rate (30.3% vs. 34.2%) and better clinical improvement (52.8% vs. 47.4%) on day 7 after starting empirical antibiotic therapy, there was no significant difference. A causative organism was isolated from BCs in 30% of septic patients, with gram-negative bacteria (GNB) predominating in 60% of cases, and multidrug-resistant (MDR) or extensively drug-resistant (XDR) bacteria predominantly detected in the BCs of the IEAT group. Although the AEAT group had slightly worse clinical characteristics at the onset of sepsis than the IEAT group, the AEAT group showed faster improvement on days 7 and 14 of sepsis. In this retrospective cross-sectional study, the AEAT group was associated with better clinical response at day 7 after sepsis onset and lower mortality, but without a significant difference. Comorbidities and the type of bacterial pathogen should also be taken into account as they can also contribute to the prediction of the final outcome. These results demonstrate the importance of daily assessment of clinical factors to more accurately predict the clinical outcome of a septic patient.

## 1. Introduction

Infections and sepsis are among the most important conditions in critically ill patients, and prompt empirical antibiotic therapy is a key component in the treatment of bacterial sepsis. Therefore, the appropriate choice of antibiotic spectrum is associated with a better outcome in patients with sepsis and septic shock [1]. Broad-spectrum antibiotics are often used as empirical therapy for the treatment of critically ill patients in the intensive care unit (ICU) [2,3]. They are defined as therapeutics with sufficient activity to cover a range of gram-negative (GNB) and gram-positive bacteria (GPB) (e.g., carbapenems and piperacillin-tazobactam). The correct choice of empirical antimicrobial therapy based on clinical and epidemiological criteria is crucial for the patient’s outcome. Empirical therapy for patients with sepsis should target the most common organisms causing sepsis in specific patient populations. The most common organisms isolated in patients with sepsis include *Escherichia coli*, *Staphylococcus aureus*, *Klebsiella pneumoniae*, and *Streptococcus pneumoniae*, so coverage of these organisms should be considered when choosing an agent [4]. One particular problem is the unstoppable increase in antimicrobial resistance. An international multicentric study analyzing the profiles of multidrug-resistant (MDR) organisms in septic ICU patients showed that one-third of the MDR organisms were *K. pneumoniae*, while more than half of the extensively drug-resistant (XDR) isolates were *Acinetobacter baumannii* [5]. Due to the high prevalence of invasive procedures and devices, comorbidity, and older age, ICU patients are at high risk of acquiring healthcare-associated infections or hospital-acquired infections (HAIs), which are often caused by MDR pathogens [6].

One of the most effective ways to combat antimicrobial resistance is to reduce the use of antibiotics. The implementation of an antibiotic-stewardship program (ASP) can reduce the use of antibiotics and antimicrobial resistance in the ICU. However, there are differences between countries in terms of ASPs and infection prevention and control programs in ICUs [7].

In view of this, the question of which type of antimicrobial agent is suitable is still controversial. The appropriate antibiotics are defined as antibiotics to which the causative pathogens are susceptible in vitro or which are appropriate for the suspected site of infection in culture-negative sepsis [8,9,10,11]. In patients at high risk of MDR organisms, current guidelines recommend the use of two agents effective against gram-negative bacteria for empirical treatment to increase the likelihood of adequate coverage [12]. Initial combination therapy usually consists of a β-lactam (third-generation cephalosporins, β-lactam/β-lactamase inhibitor agents, or carbapenems) plus an aminoglycoside or a fluoroquinolone. In patients at low risk of MDR organisms, a single agent such as piperacillin/tazobactam, an antipseudomonal cephalosporin, or carbapenem is recommended for empirical treatment.

It is, therefore, important to understand the clinical and epidemiological data and to individualize the choice of empirical therapy. Instead of universal broad-spectrum antibiotics, the specific antibiotic regimen should be determined using a more targeted approach. This approach should include site-specific diagnostics including blood cultures (BCs), the use of rapid methods for etiological diagnosis such as polymerase chain reaction (PCR) and matrix-assisted laser desorption/ionization time-of-flight mass spectrometry (MALDI-TOF). In addition, it is necessary to identify the likely pathogen based on epidemiological factors and host risk factors, assess the severity of the disease (sepsis with stable blood pressure versus septic shock), determine the likely site of infection, characterize the likelihood of MDR or XDR infection, and weigh the consequences of not using an active regimen either immediately or ultimately at the time of initial empirical selection. Patients’ risk factors include recent infections, evidence of relevant colonization, comorbidities, implanted devices, immunological status, recent infections, and antibiotic use in the last 3 months [13,14,15,16,17,18]. Several guidelines, including those of the Infectious Diseases Society of America (IDSA) [2] and local guidelines for the use of antimicrobial drugs in hospitals [3], provide answers to many of these questions. Once empirical therapy has been started, daily assessment and clinical evaluation of its effect in septic patients is required. Previous findings suggest that a good clinical response in critically ill patients with a bacterial infection within the first 7 days of starting antimicrobial treatment may be associated with a shorter duration of treatment and a more favorable outcome [19]. Accordingly, it is of great importance to identify the pathogen as quickly as possible. If this is not possible, the clinical response and antibiotic therapy must be reviewed daily.

However, the relationship between appropriate empirical antibiotic therapy, defined according to the susceptibility of the pathogen detected in blood cultures (BCs) after therapy, has already been initiated, and the clinical response a few days after a sepsis diagnosis has been insufficiently studied. As the current Surviving Sepsis Campaign guideline makes the general recommendation that 7 to 10 days of antibiotic treatment is likely to be sufficient for most severe infections associated with sepsis and septic shock, we defined day 7 as the day on which we estimate the clinical response to empirical therapy and the mortality rate [12]. We aimed to compare the clinical response and mortality on day 7 between two groups of critically ill patients with sepsis and/or septic shock: those who received adequate empirical therapy during the 28-day follow-up period, with those who did not, according to the in vitro sensitivity of bacteria detected in BCs in a single center in Croatia.

## 2. Results

During the study period, there were 510 cases of sepsis, of which 92 were excluded, leaving 418 cases in the analysis (Figure 1). Of the 418 cases, 127 patients (31%) had a positive BC, and 291 (69%) had a negative BC. Of the 127 patients with a positive BC, 89 patients (70%) received AEAT, and 38 patients (30%) received IEAT.

Baseline characteristics were generally balanced between the AEAT and IEAT groups (age, gender, comorbidities, and length of ICU stay) (Table 1), with no statistically significant differences. The mean age of the AEAT and IEAT groups was 68.9 ± 11.5 and 67.6 ± 11.7 years, respectively. There were more male patients in both groups and the length of stay in the ICU was longer in the IEAT group (median 9 vs. 5). In addition, there were significantly more patients in the IEAT group with a worse 10-year survival rate according to the CCI (*p* = 0.009), i.e., 27 out of 38 patients (71.1%) had an estimated 10-year survival rate of 53% and lower, unlike the AEAT group (41/89, 46.1%). However, the overall CCI scores showed no statistically significant difference (*p* = 0.142) between the groups. Both groups had elevated acute inflammatory factors (CRP and PCT) on the day of sepsis diagnosis, with significantly higher values in the AEAT group (*p* = 0.004 and *p* = 0.030). However, in contrast to the IEAT group, patients in the AEAT group showed a more rapid decline in these two factors and in APACHE II scores towards days 7 and 14.

Baseline characteristics were also balanced between the surviving and deceased groups (age, gender and comorbidities, length of ICU stay) (Table 2), with no statistically significant differences. The mean age of the surviving and deceased groups was 68.2 ± 13.2 and 71.3 ± 11.3 years, respectively. There were also more male patients in both groups, and there were significantly more patients with cardiovascular diseases in the deceased group (67.1% vs. 57.3%, *p* = 0.035) than in the surviving group. As expected, the deceased group had significantly higher APACHE II scores on the day of sepsis diagnosis (23.9 ± 6.7 vs. 19.2 ± 7.4, *p* = 0.000). In addition, the deceased group had significantly higher levels of PCT on day 14 than the surviving group (median 2.3 vs. 0.4, *p* = 0.030).

As shown in Table 3, there were significantly more patients in the deceased group who received inappropriate empirical antibiotic therapy than in the surviving group (65.3% vs. 23.1%, *p* = 0.000). In addition, there were more BSIs caused by ESBL-producing *Enterobacterales* in the deceased group, albeit at the margin of statistical significance (20.0% vs. 7.7%, *p* = 0.055). The group of deceased patients had more HAIs, but without a statistically significant difference (30.7% vs. 21.2%, *p* = 0.233).

The AEAT group had a better clinical outcome, i.e., more patients with an improved clinical response on day 7 after the onset of sepsis and initiation of empirical antibiotic therapy (Table 4), but without a significant difference (52.8% vs. 47.4%, *p* = 0.574). In the IEAT group, more GNB were detected in the BCs, but without a statistically significant difference (65.8% vs. 58.4%, *p* = 0.436). However, significantly more ESBL-producing *Enterobacterales* (13/38 vs. 6/89, *p* = 0.000), carbapenem-resistant *Enterobacterales* (CRE) (3/38 vs. 1/89, *p* = 0.045) and carbapenem-resistant *Pseudomonas aeruginosa* (2/38 vs. 0/89, *p* = 0.029) were detected in the BCs in the IEAT group than in the AEAT group. Other resistant bacteria, such as carbapenem-resistant *A. baumannii* (CRAB), methicillin-resistant *S. aureus* (MRSA), and vancomycin-resistant *Enterococcus* (VRE), were detected at a similar frequency in both groups (AEAT and IEAT). HAIs were detected in 26.8% of patients with positive BCs, 34.2% of whom were in the IEAT group.

The 7-day, 14-day, 28-day, and ICU mortality rates were lower in the AEAT than in the IEAT group, but no statistically significant difference was detected (Table 5).

## 3. Discussion

Appropriate empirical antibiotic therapy is considered one of the cornerstones of sepsis treatment, as it has been shown to be associated with improved survival rates [10,12]. Our results showed that the group of septic patients with appropriate initial empirical antibiotic therapy had better improvement and lower mortality, although this was not statistically significant. While the majority of patients received AEAT, slightly more than a third did not. This statistically insignificant result could simply be due to the small number of patients in the IEAT group. However, we believe that these results are of clinical significance and should not be overlooked.

According to the results of a systematic review and meta-analysis, the 30-day mortality rate for sepsis in developed countries (Europe, North America, and Australia) is 24%, and for septic shock, it is 34% [20]. In our study, a 28-day mortality rate of 50% was observed for AEAT and 60% for the IEAT group. Although another Croatian study has shown similar results to ours [21], this is a limitation that needs to be considered as the results cannot be extrapolated to other countries with different socioeconomic levels or healthcare systems.

According to the current Surviving Sepsis Campaign guideline, 7 to 10 days of antibiotic treatment is likely to be sufficient for most severe infections associated with sepsis and septic shock [12]. The current study suggests that the assessment of the clinical response in septic ICU patients, including the APACHE II score at day 7, can predict mortality. If the 7-day mortality in the AEAT group was 30.3% and a positive clinical response to the initial empirical antibiotic therapy is observed, these patients could be said to have a survival chance of about 70%. The most recent study investigated 7-day mortality and the clinical response to empirical antibiotic therapy and came to similar conclusions [19]. However, it included ICU patients with all infections, not just sepsis and/or septic shock, and analyzed them according to the clinical response rather than the BC results. In our study, the AEAT group showed faster improvement in clinical status, as measured by APACHE II scores as well as by acute inflammatory factors (CRP and PCT), than the IEAT group, although the AEAT group had higher baseline CRP and PCT levels. These findings definitely imply the importance of daily assessment of clinical factors to more accurately predict the clinical outcome of a septic patient.

As underlying chronic conditions can affect outcomes in patients with sepsis, assessment of pre-existing comorbidities is important to improve the prediction of mortality in sepsis [22]. According to previous reports, 55.5% to 65% of patients with sepsis had underlying comorbidities [23].

In a couple of recent studies, neither BC positivity nor the time to positivity was associated with 90-day mortality in adults presenting to the emergency department with severe manifestations of sepsis [24,25]. In a multicenter study, mortality was more associated with age, higher CCI, and positive qSOFA [24], similar to our single-center study. However, this multicenter study did not compare the appropriateness of empirical therapy or examine the bacterial species detected for mortality risk, and additionally, our study did not compare patients with positive and negative BC results. As mentioned, in our study the group of deceased patients was slightly older and had higher CCI and SOFA scores, but again, there was no statistically significant difference compared to the survivors. On the other hand, in a study by Yang et al., patients with a positive BC had an increased late mortality, i.e., a 28- to 90-day mortality in contrast to patients with a negative BC [26]. These findings suggest that all other patients’ risk factors and possible sources of infection should also be taken into account when treating sepsis, as BC results alone do not appear to have a significant impact on mortality in this patient population. However, in our study, no statistically significant difference in comorbidities was found in the two main groups analyzed (AEAT and IEAT), although the IEAT group had significantly more patients with an estimated 10-year survival rate of less than 53% according to the CCI. We assume this may have a negative prognostic value for the final outcome. However, we did not adjust for this factor and could not judge the association between the evaluation of the clinical response to empirical antibiotic therapy at day 7 and the 10-year survival rate. Further studies with a larger number of patients are needed to investigate this association.

The prevalence of HAIs varies between countries, hospitals, and different hospital departments. The ICU represents a special environment where HAIs are acquired to a greater extent [27]. The ECDC prevalence study carried out in European acute care hospitals from 2022 to 2023 reported the highest prevalence of HAIs in patients admitted to intensive care (20.5%). BSIs were one of the most common types of HAIs in the ICU [28]. Similar to another study from Eastern Europe, we found a very high rate of HAI in our ICU, about one-third of BC-positive patients acquired a HAI [29]. Comparing patients with HAIs and CAIs, we found that the rate of HAIs was higher in the deceased group, which probably contributed to the increased mortality. This is not surprising considering that they are mostly caused by resistant bacteria. In fact, we calculated that 18/25 HAIs in the deceased group were caused by MDR bacteria (72.0%), of which 9/18 (50.0%) were GNB and mainly CRAB 6/9 (66.7%).

There is still a controversy about which bacteria are more harmful, gram-positive or gram-negative bacteria. Some suggest that an infection with GNB causes a stronger inflammatory reaction of the host [30], others that there is no significant difference in the prognosis of sepsis caused by GNB and GPB [31]. More recent results of a meta-analysis study showed that there is no significant difference in survival or length of hospital stay [32]. However, the incidence of severe sepsis and serum levels of inflammatory factors (CRP, PCT, TNF-α) were higher in gram-negative sepsis than in gram-positive sepsis. In our study, although not statistically significant, the IEAT group had more BCs with proven GNB, which could be explained by the higher local susceptibility of gram-positive bacteria with a low prevalence of MDR/XDR isolates (VRE and MRSA). Moreover, significantly more ESBL-producing *Enterobacterales* and other highly resistant GNB (CRE, CRPA) were detected in the IEAT group.

The choice of adequate empirical antibiotic therapy for gram-negative sepsis becomes more difficult when considering that the resistance of gram-negative bacteria to carbapenems is increasing locally and globally. During the study period, the initial empirical regimen used in our ICU consisted mainly of ceftriaxone or piperacillin-tazobactam. The antibiotic combination therapy routinely used was vancomycin with ceftriaxone or meropenem. If carbapenem-resistant *Acinetobacter* was suspected, colistin was added to meropenem. In most cases, the reason for the inappropriateness of therapy was the lack of specific coverage for ESBL-producing *Enterobacterales* and other highly resistant GNBs (CRE, CRAB, CRPA).

Our study has several limitations: First, the results of our study are limited by the small sample size and low statistical power, which is a common limitation of retrospective studies. In addition, we were unable to perform a multivariable analysis of mortality due to insufficient power. Secondly, an unbalanced BC-positivity rate between the AEAT and IEAT groups may have biased the analyses. Third, we did not adjust the studied groups (AEAT, IEAT, survivors, deceased) for CCIs and clinical factors (APACHE II, CRP, PCT) on the day of sepsis diagnosis and did not exclude those in whom antibiotic therapy was replaced before day 7, nor those who received the antibiotic therapy before the onset of sepsis in the previous department for another reason, i.e., prophylaxis or possible local infection, as these are subject to numerous variations and cannot be accurately assessed in a study with a relatively small number of participants. Fourth, other elements that contribute to clinical outcomes were also not considered, such as the adequacy of the control of the infection source and the time to targeted therapy. Large prospective studies are needed to confirm these results.

## 4. Material and Methods

### 4.1. Study Design, Data Collection, and Definitions

This retrospective cohort study was conducted from January 2020 to December 2023 in western Croatia at the Clinical Hospital Center Rijeka with 1069 general beds and 27 beds in the ICU. The clinical data were extracted from the patients’ medical records using the hospital’s computerized and paper archival databases. The following data were collected: demographic data (age, gender), comorbidities, Sequential Organ Failure Assessment (SOFA) score on the day of sepsis diagnosis, Acute Physiology And Chronic Health Evaluated II (APACHE II) score on the day of sepsis diagnosis, on day 7, and on day 14 after sepsis diagnosis, empirical antibiotic therapy, BC results, presence of MDR/XDR bacteria, clinical response to the initial empirical therapy on day 7 after sepsis diagnosis, length of ICU stay, and clinical outcomes.

The study was conducted in accordance with the ethical principles of the Declaration of Helsinki and the current guidelines for good clinical practice. The local ethics committee approved the study. As we analyzed anonymous data, written informed consent was not required.

The Charlson Comorbidity Index (CCI) was used as an aggregate measure of comorbidities [33]. Additionally, patients were stratified according to their CCI score into 3 categories, 0–1 point, 2–3 points, and ≥4, to better differentiate the level of the estimated 10-year survival rate. Empirical antibiotic therapy was defined as treatment with antibiotics started either on the day the BCs were collected or the day after.

Empirical antimicrobial agents were selected based on clinical judgment by infectious disease specialists, which could subsequently be changed depending on the clinical response over the next few days after the sepsis diagnosis and depending on the final BC results: in patients with severe community-acquired pneumonia without risk factors for a pseudomonal infection, a combination of ceftriaxone with macrolides or fluoroquinolones was used. Carbapenems (mostly meropenem, less frequently imipenem) were used as empirical therapy in most patients with possible extended-spectrum β-lactamase (ESBL)-producing *Enterobacterales* infections (e.g., long hospitalization, placement of urinary catheters, recent antimicrobial use). Vancomycin was added to patients with risk factors for methicillin-resistant *S. aureus* (MRSA) infections such as a known MRSA colonization or a previous infection with MRSA, a stay in a long-term care facility, or a previous intensive exposure to antibiotics. Piperacillin-tazobactam was used in patients at risk of pseudomonal infection (e.g., chronic lung disease, previous antibiotic therapy) and colistin in patients previously colonized with XDR *Acinetobacter*.

Contamination was considered to be the presence of coagulase-negative staphylococci (CoNS), *Bacillus,* or *Corynebacterium* species isolated from only one of at least two sets of BCs. Empirical antibiotic therapy was considered appropriate (AEAT) if the bacteria detected *in vitro* showed sensitivity to at least one antibiotic administered. Empirical antibiotic therapy was considered inappropriate (IEAT) if the isolate from the bloodstream showed no sensitivity. MDR was defined as acquired non-susceptibility to at least one agent in three or more antimicrobial categories, XDR was defined as non-susceptibility to at least one agent in all but two or fewer antimicrobial categories (i.e., bacterial isolates remain susceptible to only one or two categories) [34].

HAI was defined as an infection that occurred during healthcare in a hospital or other healthcare facility and that first occurred 48 h or more after hospital admission or within 30 days of healthcare. The HAI was identified using the European Center for Disease Control (ECDC) criteria [28]. A community-acquired infection (CAI) was defined as an infection acquired outside of a healthcare facility or an infection present at the time of admission.

At the onset of sepsis, and on days 7 and 14, the severity of the disease was assessed by SOFA and APACHE II scores and two factors of the acute inflammatory response, i.e., C-reactive protein (CRP) and procalcitonin (PCT). The clinical response or efficacy of the empirical antibiotic therapy initiated was defined as therapy that clinically improved or cured the patient. Clinical response was assessed using the APACHE II score and categorized into two groups: as clinically cured or improved on day 7 and as clinically deteriorated on day 7.

### 4.2. Diagnosis of Bloodstream Infection and Microbiological Tests

The automated blood culture system BacT/Alert Virtuo (bioMérieux, Marcy l’Etoile, France) was used to detect microorganisms in the blood. Positive bottles were processed according to the laboratory’s standard operating procedures. The bottles were subcultured onto a solid medium for further analyses, including identification using the automated Vitek 2 system (bioMe’rieux, Marcy l’Etoile, France) and the Biofire Filmarray Multiplex PCR system (bioMe’rieux, Marcy l’Etoile, France). Antibiotic susceptibility was assessed according to the current standard of the European Committee on Antimicrobial Susceptibility Testing (EUCAST) [35].

### 4.3. Participants

ICU patients with sepsis and/or septic shock aged ≥ 18 years with initiated empirical antimicrobial therapy were included in the study. Sepsis was defined according to Sepsis-3 clinical criteria as an acute increase in the SOFA score ≥ 2 associated with a suspected or documented infection [36]. This scoring system as a simplified version of the SOFA (also known as quickSOFA, qSOFA) uses 3 criteria, assigning one point for low blood pressure (systolic blood pressure ≤ 100 mmHg), high respiratory rate (≥22 breaths per min) or altered mental status (Glasgow Coma Scale < 15). In addition, to determine the degree of organ dysfunction and mortality risk in ICU patients on admission, the SOFA scoring system was used based on laboratory results and clinical data, which evaluates the performance of multiple organ systems (neurologic, blood, liver, kidney, and blood pressure/hemodynamics) and assigns a score based on the data obtained in each category.

Each patient was included only once during hospitalization. Patients with missing data, with contaminated BC, with fungal sepsis, and with diagnosed COVID-19 and/or influenza were excluded from the study.

Depending on the BC results, we categorized the patients into two groups: BC-negative and BC-positive. Based on the antibiotic sensitivity of the bacteria isolated from the BC, we further divided the patients with positive BCs into the group that received appropriate therapy (the AEAT group) and the group that received inappropriate empirical therapy (the IEAT group). Patients with isolated bacteria from other sources of infection, but a negative BC were assigned to the negative BC group. We also divided the patients into the surviving group and the deceased group according to the final clinical outcome and analyzed the appropriateness of the empirical antibiotic therapy and the microbiological profile of the BSI.

### 4.4. Outcome Measures

Our primary outcomes were to compare the clinical response (APACHE II, CRP, and PCT) to empirical antibiotic treatment at day 7, and 7-day mortality between two groups of septic patients—with appropriate (AEAT group) and inappropriate (IEAT group) empirical antibiotic therapy.

The secondary outcomes were the comparison of baseline demographic and clinical data, the type of BSI (gram-positive or gram-negative), the origin of the BSI (HAI or CAI), and the mortality rate (14-day, 28-day, and ICU mortality) between the AEAT and IEAT groups of patients. In addition, the secondary outcomes were also to compare demographic and clinical characteristics, the type and origin of the BSI, and furthermore, the appropriateness of the antimicrobial therapy between the surviving and deceased group of patients.

### 4.5. Statistical Analysis

Data were analyzed using Statistica, version 13 (TIBCO Software Inc., 2017, Palo Alto, CA, USA). Data distribution was tested for normality using the Kolmogorov–Smirnov test. We compared categorical variables using the Chi-square test, and continuous variables using the Student’s *t*-test or Mann–Whitney U test, as appropriate. Continuous variables are presented as medians or means, and categorical variables are expressed as numbers and percentages. The statistical significance threshold was set at *p* < 0.05.

## 5. Conclusions

In this retrospective cross-sectional study, appropriate initial empirical antimicrobial therapy was not associated with a statistically significantly better clinical response on day 7 after the onset of sepsis and lower mortality. Comorbidities should also be considered, especially by estimating the 10-year survival rate, in order to make a more accurate prediction.

In addition, the type of bacterial pathogen can also contribute to the prediction of the final outcome, which is why it is important to detect it as early as possible. However, if this is not possible, i.e., if the BCs are negative or other microbiological methods show negative results, the infection sites and/or possible MDR/XDR pathogens (e.g., CRE, CRAB, CRPA, VRE, MRSA) should be considered and covered accordingly. These findings imply the importance of daily assessment of clinical factors to predict the clinical outcome of a septic patient more accurately.

A large randomized controlled trial is needed to confirm our results in specific subgroups.

## Figures and Tables

**Figure 1 antibiotics-13-00569-f001:**
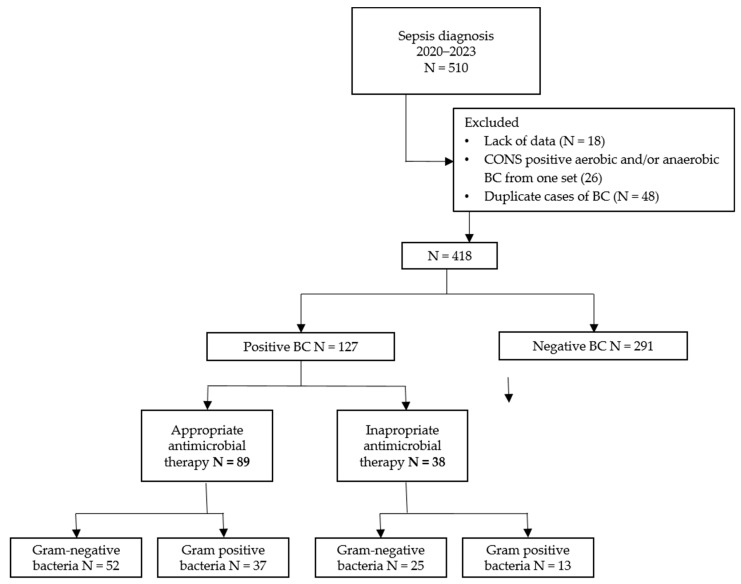
Flowchart of the number of participants in the different phases of the study. Abbreviations: *n*, number; BC, blood culture.

**Table 1 antibiotics-13-00569-t001:** Clinical-demographic characteristics of AEAT and IEAT groups of patients in ICU on the day of onset of sepsis/initiation of empirical antibiotic therapy, and on the 7th and 14th day after.

Variables	BC-Positive Group (*n* = 127/418, 31%)	AEAT Group (*n* = 89/127, 70%)	IEAT Group (*n* = 38/127, 30%)	*p*
^a^ Age, mean ± SD	68.6 ± 11.5	68.9 ± 11.5	67.6 ± 11.7	0.545
^b^ Female, *n* (%)	48/127 (37.8)	32/89 (35.9)	16/38 (42.1)	0.512
^b^ Comorbidities, *n* (%)				
Cardiovascular disease	76/127 (59.8)	54/89 (60.7)	22/38 (57.9)	0.769
Diabetes mellitus	24/127 (18.9)	14/89 (15.7)	10/38 (26.3)	0.162
Solid tumor	7/127 (5.5)	6/89 (6.7)	1/38 (2.6)	0.352
Cerebrovascular disease	12/127 (9.4)	6/89 (6.7)	6/38 (15.8)	0.110
Chronic renal failure	13/127 (10.2)	8/89 (9.0)	5/38 (13.2)	0.477
Chronic pulmonary disease	19/127 (14.9)	15/89 (16.9)	4/38 (10.5)	0.359
Chronic hepatic disease	12/127 (9.4)	6/89 (6.7)	6/38 (15.8)	0.110
Hematologic malignancy	8/127 (6.3)	7/89 (7.9)	1/38 (2.6)	0.266
No chronic illness	20/127 (15.7)	13/89 (14.6)	7/38 (18.4)	0.588
Immunocompromised host	10/127 (7.9)	10/89 (11.2)	0/38 (0.0)	0.107
CCI				
^b^ CCI categories, *n* (%)				
CCI 0–1	10/127 (7.9)	6/89 (6.7)	4/38 (10.5)	0.468
CCI 2–3	49/127 (38.6)	42/89 (47.2)	7/38 (18.4)	0.002 *
CCI ≥ 4	68/127 (53.5)	41/89 (46.1)	27/38 (71.1)	0.009 *
^c^ CCI scores, median (range)	4 (0–10)	3 (0–10)	4 (1–10)	0.142
^c^ Lenght of ICU stay (days), median (range)	6.0 (1.0–42.0)	5.0 (1.0–42.0)	9.0 (1.0–37.0)	0.186
Factors on day of sepsis onset/initiation of empirical antibiotic therapy, mean ± SD or median (range)				
^a^ SOFA	9.8 ± 3.8	10.1 ± 3.6	9.1 ± 9.3	0.240
^a^ APACHE II	20.9 ± 7.7	20.6 ± 7.8	21.7 ± 7.6	0.957
^a^ CRP (mg/L)	195.5 ± 116.2	215.2 ± 119.9	148.4 ± 93.3	0.004 *
^c^ PCT (ng/mL)	12.8 (0.09–149.0)	14.8 (0.09–118.0)	2.4 (0.1–149.0)	0.030 *
Factors on day 7, mean ± SD or median (range)				
^a^ APACHE II	17.2 ± 6.7	17.0 ± 7.3	17.5 ± 5.9	0.580
^a^ CRP (mg/L)	134.3 ± 85.0	127.1 ± 73.2	144.0 ± 99.7	0.465
^c^ PCT (ng/mL)	1.5 (0.1–137.0)	2.4 (0.1–120.0)	0.6 (0.1–137.0)	0.092
Factors on day 14, mean ± SD or median (range)				
^a^ APACHE II	15.0 ± 7.3	13.6 ± 6.4	16.3 ± 8.1	0.564
^a^ CRP (mg/L)	143.9 ± 89.5	129.3 ± 81.4	157.4 ± 97.9	0.408
^c^ PCT (ng/mL)	0.9 (0.1–8.6)	0.9 (0.1–7.9)	1.0 (0.3–8.6)	0.772

Abbreviations: AEAT, appropriate empirical antibiotic therapy; IEAT, inappropriate empirical antibiotic therapy; *n*, number; BC, blood culture; SD, standard deviation; * statistical significance; CCI, Charlson Comorbidity Index; ICU, intensive care unit; SOFA, sequential organ failure assessment; APACHE II, acute physiology and chronic health evaluation II; CRP, c-reactive protein; PCT, procalcitonin. ^a^ Student’s *t*-test; ^b^ Chi-square test; ^c^ Mann–Whitney U test.

**Table 2 antibiotics-13-00569-t002:** Clinical-demographic characteristics of surviving and deceased patients in ICU on the day of onset of sepsis/initiation of empirical antibiotic therapy, and on the 7th and 14th day after.

Variables	All Patients (*n* = 418)	Surviving Group (*n* = 184/418, 44%)	Deceased Group (*n* = 234/418, 56%)	*p*
^a^ Age, mean ± SD	69.9 ± 12.3	68.2 ± 13.2	71.3 ± 11.3	0.242
^b^ Female, *n* (%)	150/418 (35.9)	59/184 (32.1)	91/234 (38.9)	0.148
^b^ Comorbidities, *n* (%)				
Cardiovascular disease	262/418 (62.7)	105/184 (57.3)	157/234 (67.1)	0.035 *
Diabetes mellitus	86/418 (20.6)	33/184 (17.9)	53/234 (22.8)	0.236
Solid tumor	41/418 (9.8)	20/184 (10.9)	20/234 (8.7)	0.422
Cerebrovascular disease	64/418 (15.3)	23/184 (12.5)	41/234 (17.4)	0.157
Chronic renal failure	44/418 (10.5)	20/184 (11.0)	24/234 (10.1)	0.839
Chronic pulmonary disease	56/418 (13.4)	29/184 (15.7)	27/234 (11.4)	0.208
Chronic hepatic disease	42/418 (10.0)	17/184 (9.2)	25/234 (10.7)	0.625
Hematologic malignancy	19/418 (4.5)	8/184 (4.2)	11/234 (4.3)	0.863
No chronic illness	62/418 (14.8)	34/184 (18.5)	28/234 (11.9)	0.063
Immunocompromised host	31/418 (7.4)	17/184 (9.39	14/234 (5.9)	0.207
CCI				
^b^ CCI categories, *n* (%)				
CCI 0–1	31/418 (7.4)	14/184 (7.6)	16/234 (6.8)	0.761
CCI 2–3	121/418 (28.9)	62/184 (33.7)	63/234 (26.9)	0.300
CCI ≥ 4	264/418 (63.2)	109/184 (59.2)	155/234 (66.2)	0.140
^c^ CCI scores, median (range)	4 (0–11)	4 (0–10)	5 (0–11)	0.204
^c^ Lenght of ICU stay (days), median (range)	5.0 (1.0–42.0)	6.0 (1.0–32.0)	5.0 (1.0–42.0)	0.873
Factors on day of sepsis onset/initiation of empirical antibiotic therapy, mean ± SD or median (range)				
^a^ SOFA	10.3 ± 3.6	9.4 ± 3.4	11.1 ± 3.7	0.153
^a^ APACHE II	21.8 ± 7.4	19.2 ± 7.4	23.9 ± 6.7	0.000 *
^a^ CRP (mg/L)	197.9 ± 118.4	184.3.6 ± 117.1	184.1 ± 118.5	0.988
^c^ PCT (ng/mL)	35.4 (0.09–149.0)	6.6 (0.09–117.0)	11.7 (0.1–149.0)	0.312
Factors on day 7, mean ± SD or median (range)				
^a^ APACHE II	16.4 ± 6.6	14.0 ± 5.6	18.4 ± 6.6	0.197
^a^ CRP (mg/L)	132.5 ± 80.9	126.4 ± 76.9	137.1 ± 85.0	0.545
^c^ PCT (ng/mL)	10.1 (0.1–137.0)	1.3 (0.1–120.0)	1.7 (0.1–137.0)	0.137
Factors on day 14, mean ± SD or median (range)				
^a^ APACHE II	16.7 ± 7.2	14.0 ± 7.9	18.2 ± 6.3	0.882
^a^ CRP (mg/L)	140.1 ± 94.4	108.3 ± 74.4	158.2 ± 101.0	0.122
^c^ PCT (ng/mL)	5.1 (0.1–101.0)	0.4 (0.1–14.0)	2.3 (0.1–101.0)	0.030 *

Abbreviations: ICU, intensive care unit; *n*, number; SD, standard deviation; * statistical significance; CCI, Charlson Comorbidity Index; SOFA, sequential organ failure assessment; APACHE II, acute physiology and chronic health evaluation II; CRP, c-reactive protein; PCT, procalcitonin. ^a^ Student’s *t*-test; ^b^ Chi-square test; ^c^ Mann–Whitney U test.

**Table 3 antibiotics-13-00569-t003:** Appropriateness of empirical antibiotic therapy and microbiological profile of BSI in surviving and deceased patients.

Variables	Surviving Group with Positive BC (*n* = 52/127, 41%)	Deceased Group with Positive BC (*n* = 75/127, 59%)	*p*
^a^ Appropriateness of therapy, *n* (%)			
AEAT	40/52 (76.9)	26/75 (34.7)	0.000 *
IEAT	12/52 (23.1)	49/75 (65.3)	0.000 *
^a^ Type of BCI, *n* (%)			
GNB	35/52 (67.3)	42/75 (56.0)	0.199
ESBL	4/52 (7.7)	15/75 (20.0)	0.055
CRE	1/52 (1.9)	3/75 (4.0)	0.509
CRAB	2/52 (3.8)	7/75 (9.3)	0.236
CRPA	0/52 (0.0)	2/75 (2.7)	0.235
GPB	17/52 (32.7)	33/75 (44.0)	0.199
MRSA	1/52 (1.9)	5/75 (6.7)	0.215
VRE	0/52 (0.0)	1/75 (1.3)	0.403
^a^ Origin of infection, *n* (%)			
CAI	41/52 (78.8)	52/75 (69.3)	0.233
HAI	11/52 (21.2)	23/75 (30.7)	0.233

Abbreviations: BSI, bloodstream infection; *n*, number; BC, blood culture; AEAT, appropriate empirical antibiotic therapy; IEAT, inappropriate empirical antibiotic therapy; * statistical significance; GNB, gram-negative bacteria; ESBL, extended-spectrum β-lactamases; CRE, carbapenem-resistant *Enterobacterales*; CRAB, carbapenem-resistant *A. baumannii*; CRPA, carbapenem-resistant *P. aeruginosa*; GPB, gram-positive bacteria; MRSA, methicillin-resistant *S. aureus*; VRE, vancomycin-resistant *Enterococcus*; CAI, community-acquired infection; HAI, hospital-acquired infection. ^a^ Chi-square test.

**Table 4 antibiotics-13-00569-t004:** Clinical response to the initial empirical antibiotic therapy on day 7 and microbiological profile of the BSI.

Variables	BC-Positive Group (*n* = 127/418, 30%)	AEAT Group (*n* = 89/127, 70%)	IEAT Group (*n* = 38/127, 30%)	*p*
^a^ Clinical response on 7 days, *n* (%)				
Improved/Cured	65/127 (51.2)	47/89 (52.8)	18/38 (47.4)	0.574
Deteriorated	62/127 (48.8)	42/89 (47.2)	20/38 (52.6)	0.574
^a^ Type of BSI, *n* (%)				
GNB	77/127 (60.6)	52/89 (58.4)	25/38 (65.8)	0.436
ESBL	19/127 (14.9)	6/89 (6.7)	13/38 (34.2)	0.000 *
CRE	4/127 (3.1)	1/89 (1.1)	3/38 (7.9)	0.045 *
CRAB	9/127 (7.1)	4/89 (4.5)	5/38 (13.2)	0.081
CRPA	2/127 (1.6)	0/89 (0.0)	2/38 (5.3)	0.029 *
GPB	50/127 (39.8)	37/89 (41.6)	13/38 (34.2)	0.436
MRSA	6/127 (4.7)	6/89 (6.7)	0 (0.0)	0.101
VRE	1/127 (0.8)	1/89 (1.1)	0 (0.0)	0.511
Origin of infection, *n* (%)				
CAI	93/127 (73.2)	68/89 (76.4)	25/38 (65.8)	0.216
HAI	34/127 (26.8)	21/89 (23.6)	13/38 (34.2)	0.216

Abbreviations: BSI, bloodstream infection; *n*, number; BC, blood culture; AEAT, appropriate empirical antibiotic therapy; IEAT, inappropriate empirical antibiotic therapy; GNB, gram-negative bacteria; * statistical significance; ESBL, extended-spectrum β-lactamases; CRE, carbapenem-resistant *Enterobacterales*; CRAB, carbapenem-resistant *A. baumannii*; CRPA, carbapenem-resistant *P. aeruginosa*; GPB, gram-positive bacteria; MRSA, methicillin-resistant *S. aureus*; VRE, vancomycin-resistant *Enterococcus*; CAI, community-acquired infection; HAI, hospital-acquired infection. ^a^ Chi-square test.

**Table 5 antibiotics-13-00569-t005:** Mortality rate of patients during their stay in the ICU and on the 7th, 14th, and 28th day after the onset of sepsis/initiation of empirical antibiotic therapy.

Variables	All Patients (*n* = 418)	BC-Positive Group (*n* = 127/418, 30%)	AEAT Group (*n* = 89/127, 70%)	IEAT Group (*n* = 38/127, 30%)	*p*
^a^ Clinical outcome, *n* (%)					
7-day mortality	149/418 (35.6)	40/127 (31.5)	27/89 (30.3)	13/38 (34.2)	0.666
14-day mortality	198/418 (47.4)	56/127 (44.1)	38/89 (42.7)	18/38 (47.4)	0.627
28-day mortality	214/418 (51.2)	68/127 (53.5)	45/89 (50.6)	23/38 (60.5)	0.302
ICU-mortality	234/418 (55.9)	75/127 (59.1)	49/89 (55.1)	26/38 (68.4)	0.160

Abbreviations: *n*, number; BC, blood culture; AEAT, appropriate empirical antibiotic therapy; IEAT, inappropriate empirical antibiotic therapy; ICU, intensive care unit. ^a^ Chi-square test.

## Data Availability

Data supporting this article is available from the corresponding author upon reasonable request.

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
