# Peer review of "Impact of Appropriate Empirical Antibiotic Treatment on the Clinical Response of Septic Patients in Intensive Care Unit: A Single-Center Observational Study"

_antibiotics, 2024, doi:10.3390/antibiotics13060569_

Round 1
Reviewer 1 Report
Comments and Suggestions for Authors
The original article entitled “Impact of Appropriate Empirical Antibiotic Treatment on the 2 Clinical Response of Septic Patients in Intensive Care Unit: A Single‑Center Observational Study” presented the importance of empirical antibiotic therapy in the ICU unit. This retrospective study showed that the AEAT group was associated with better clinical response at day 7 after sepsis onset and lower mortality, although it had no significant difference. It is an interesting result and requires minor revision before acceptance. Some points need clarification as follows.
1. Line 55-59; Ref 5 – 8 are missing.
2. Line 114; The authors could give an example of empirical antibiotic regimen and its use of bacterial treatment.
3. Line 128; provide the definition of SOFA score ³ 2 for more understanding this criteria.
4. Line 142; What is the primary outcome in the clinical response? The authors could give the information.
5. Line 162 and 327; I suggest using “negative BC” instead of “sterile”.
Author Response
We would like to thank the Editors and the Reviewers for their useful suggestions and the effort they made to improve our manuscript. We have taken the comments very seriously and have tried to address all issues. We hope that these changes will be satisfactory. Please find all our responses below:
Reviewer 1:
The original article entitled “Impact of Appropriate Empirical Antibiotic Treatment on the Clinical Response of Septic Patients in Intensive Care Unit: A Single‑Center Observational Study” presented the importance of empirical antibiotic therapy in the ICU unit. This retrospective study showed that the AEAT group was associated with better clinical response at day 7 after sepsis onset and lower mortality, although it had no significant difference. It is an interesting result and requires minor revision before acceptance. Some points need clarification as follows.
Reviewer 1: 1. Line 55-59; Ref 5 – 8 are missing.
Response: we thank the reviewer. We have changed the mistake and added some new references (Line 66, highlighted).
Reviewer 1: 2. Line 114: The authors could give an example of empirical antibiotic regimen and its use of bacterial treatment.
Response: We have accepted the suggestion and added an example of empirical antibiotic regimen and its use (Lines 128-136, highlighted).
Reviewer 1: 3. Line 128; provide the definition of SOFA score ³ 2 for more understanding this criteria.
Response: we thank the reviewer. We have added the explanation (Lines 168-175, highlighted).
Reviewer 1: 4. Line 142; What is the primary outcome in the clinical response? The authors could give the information.
Response: we thank the reviewer. We have added primary outcomes (Lines 189-190, highlighted).
Reviewer 1: 5. Line 162 and 327; I suggest using “negative BC” instead of “sterile”.
Response: we thank the reviewer for the good suggestion. We have made this change in text (Lines 214, 452, highlighted).

Reviewer 2 Report
Comments and Suggestions for Authors
The BC positivity rate was only 31%, with an unbalanced positivity rate between the AEAT and IEAT groups, which induces a bias in all the assessments. According to the Surviving Sepsis Campaign 2021, antibiotic treatment should be considered depending on daily assessments instead of fixed intervals, e.g., 7, 10, or 14 days.
A very good major article published by Paquette et al. concluded: “Although severe sepsis is an inflammatory condition triggered by infection, its 90-day survival is not influenced by blood culture positivity nor its time to positivity.”
In contrast with the authors, Yang et al. recently published a study, “Outcomes Between Positive and Negative Blood Culture Septic Patients,” showing a similar early survival but a major difference in late survival rates at more than 30 days. Most probably, the observed differences at 7 and 14 days are not relevant. Actually, there are more similarities between the two groups except for the size and BC positivity rates.
The study design is similar to Tanaka et al.’s research, but the significant difference in survival observed at 7 days between the groups may be related to the inclusion of positive BC and isolated bacteria from other infection foci, reaching a positivity rate of almost 50%. Also, they mentioned the change of antibiotics and de-escalation within 7 days.
No exclusion criteria or information on fungal sepsis were made.
Please include other references and comments in the discussion section. Also, the authors should reconsider their conclusions.
Paquette K, Sweet D, Stenstrom R, Stabler SN, Lawandi A, Akhter M, et al. Neither Blood Culture Positivity nor Time to Positivity Is Associated With Mortality Among Patients Presenting With Severe Manifestations of Sepsis: The FABLED Cohort Study. Open Forum Infect Dis. 2021 Jun 17;8(7):ofab321. doi: 10.1093/ofid/ofab321. Yang L, Lin Y, Wang J, Song J, Wei B, Zhang X, Yang J, Liu B. Comparison of Clinical Characteristics and Outcomes Between Positive and Negative Blood Culture Septic Patients: A Retrospective Cohort Study. Infect Drug Resist. 2021 Oct 12;14:4191-4205. doi: 10.2147/IDR.S334161.Kim JS, Kim YJ, Kim WY. Characteristics and clinical outcomes of culture-negative and culture-positive septic shock: a single-center retrospective cohort study. Crit Care. 2021 Jan 6;25(1):11. doi: 10.1186/s13054-020-03421-4.
Comments on the Quality of English Language
Please check the grammar and consider rephrasing.
line 66 Patients risks / Patients' risk factors
line 109 was / were
.........
Author Response
We would like to thank the Editors and the Reviewers for their useful suggestions and the effort they made to improve our manuscript. We have taken the comments very seriously and have tried to address all issues. We hope that these changes will be satisfactory. Please find all our responses below:
Reviewer 2:
Reviewer 2: The BC positivity rate was only 31%, with an unbalanced positivity rate between the AEAT and IEAT groups, which induces a bias in all the assessments. According to the Surviving Sepsis Campaign 2021, antibiotic treatment should be considered depending on daily assessments instead of fixed intervals, e.g., 7, 10, or 14 days.
Response: we thank the reviewer for this remark. Our intention was to analyze a point that can tell us about the effectiveness of the empirical therapy that has been initiated. We are aware that empirical therapy must and should be evaluated on a daily basis. However, we did not include the analysis of clinical data before this day or between days 7 and 14. This requires a larger study that includes daily assessments of clinical data. We also pointed this out in the discussion (lines 270-272). We have now also added this sentence in the conclusion part (Lines 454-456, highlighted).
We are also aware of the possible bias between the groups. We have added this limitation in limitation part (Lines 431-432, highlighted).
Reviewer 2:
- A very good major article published by Paquette et al. concluded: “Although severe sepsis is an inflammatory condition triggered by infection, its 90-day survival is not influenced by blood culture positivity nor its time to positivity.”
- In contrast with the authors, Yang et al. recently published a study, “Outcomes Between Positive and Negative Blood Culture Septic Patients,” showing a similar early survival but a major difference in late survival rates at more than 30 days. Most probably, the observed differences at 7 and 14 days are not relevant. Actually, there are more similarities between the two groups except for the size and BC positivity rates.
Response: Our aim was not to compare outcomes between patients with positive and negative BC, but to compare the clinical response to empirical antibiotic treatment and mortality between patients with appropriate and inappropriate empirical therapy. We have tried to emphasize the importance of daily assessment of antibiotic therapy to consider earlier modification of initiated therapy depending on other data (comorbidities, earlier hospitalizations, other infection sites, molecular results and/or risks for MDR/XDR infections), not only depending on BCs. However, we have added further discussion regarding this thematic (Lines 374-387, highlighted).
Reviewer 2: The study design is similar to Tanaka et al.’s research, but the significant difference in survival observed at 7 days between the groups may be related to the inclusion of positive BC and isolated bacteria from other infection foci, reaching a positivity rate of almost 50%. Also, they mentioned the change of antibiotics and de-escalation within 7 days.
Response: we thank the reviewer for noticing that difference. Yes, indeed, we agree with the reviewer. However, as we added in lines 183-184, we assigned patients with isolated bacteria from other sources of infection, to the negative BC group. In future studies we will include all known foci and perform a different approach and a more detailed analysis. We did not exclude or separately analyzed patients with changed antibiotic therapy or de-escalation within these 7 days as the number of patients was too small to compare (explanation in Limitation part, Lines 437-438).
Reviewer 2: No exclusion criteria or information on fungal sepsis were made.
Response: we thank the reviewer for this observation. We added a short explanation in exclusion part. (Line 177, highlighted.)
Reviewer 2: Please include other references and comments in the discussion section. Also, the authors should reconsider their conclusions.
- Paquette K, Sweet D, Stenstrom R, Stabler SN, Lawandi A, Akhter M, et al. Neither Blood Culture Positivity nor Time to Positivity Is Associated With Mortality Among Patients Presenting With Severe Manifestations of Sepsis: The FABLED Cohort Study. Open Forum Infect Dis. 2021 Jun 17;8(7):ofab321. doi: 10.1093/ofid/ofab321.
- Yang L, Lin Y, Wang J, Song J, Wei B, Zhang X, Yang J, Liu B. Comparison of Clinical Characteristics and Outcomes Between Positive and Negative Blood Culture Septic Patients: A Retrospective Cohort Study. Infect Drug Resist. 2021 Oct 12;14:4191-4205. doi: 10.2147/IDR.S334161.
- Kim JS, Kim YJ, Kim WY. Characteristics and clinical outcomes of culture-negative and culture-positive septic shock: a single-center retrospective cohort study. Crit Care. 2021 Jan 6;25(1):11. doi: 10.1186/s13054-020-03421-4.
Response: we thank the reviewer for suggestions. We added more discussion by including these references in the text and in the reference list (highlighted). In the Conclusion part we added a relevant sentence (Lines 454-456, highlighted.)
Reviewer 2: Comments on the Quality of English Language
Please check the grammar and consider rephrasing.
line 66 Patients risks / Patients' risk factors
line 109 was / were
Response: we thank the reviewer for noticing mistakes: We have made changes (Lines 81 and 124, highlighted).

Reviewer 3 Report
Comments and Suggestions for Authors
I have read with interest the manuscript submitted by Ticac et al, since this topic represents a major concern.
I have a few comments to be addressed in order to improve the quality of the manuscript:
- introduction - I suggest adding the current recommendations for the empirical antibiotic treatment in sepsis
- rows 53-55 - I suggest adding information from these multicentric studies: https://doi.org/10.1016/j.jiph.2022.07.009 and https://pubmed.ncbi.nlm.nih.gov/34155547/which focus on this topic
- row 61 - it is worth mentioning the more rapid methods for the etiological diagnosis (e.g. molecular)
- the introduction should include the aim of the study
- in the introduction/material and methods, it is worth including the definition of sepsis or MDR/XDR
- in the material and methods section it should be included further data on the microbiological methods used
- any information about the type of infection (eg community-acquired vds hospital-acquired and the impact on the outcome/etiology?)
- unfortunately, I am not able to view the supplementary files. I suggest adding the tables in the main manuscript, if they are not too big (the manuscript in its current form is not too long and could include further information).
- the reference list is scarce and not edited according to the mdpi pattern.
Overall the manuscript is well-written/documented and could be published after some adjustments.
Best regards,
The reviewer
Author Response
We would like to thank the Editors and the Reviewers for their useful suggestions and the effort they made to improve our manuscript. We have taken the comments very seriously and have tried to address all issues. We hope that these changes will be satisfactory. Please find all our responses below:
Reviewer 3:
Reviewer 3: I have read with interest the manuscript submitted by Ticac et al, since this topic represents a major concern.
I have a few comments to be addressed in order to improve the quality of the manuscript:
Reviewer 3: - introduction - I suggest adding the current recommendations for the empirical antibiotic treatment in sepsis
Response: We thank the reviewer for his comment. We have added the current recommendations for empirical antibiotic treatment in sepsis (Lines 55-71, highlighted).
Reviewer 3: - rows 53-55 - I suggest adding information from these multicentric studies:
https://doi.org/10.1016/j.jiph.2022.07.009 and https://pubmed.ncbi.nlm.nih.gov/34155547/ which focus on this topic
Response: We thank the reviewer. We have added information from the multicentric study https://pubmed.ncbi.nlm.nih.gov/34155547/ (Lines 55-60, highlighted). However, another proposed study https://doi.org/10.1016/j.jiph.2022.07.009 analyzes and compares core antimicrobial resistance prevention measures (antibiotic stewardship and infection prevention and control programs) in different ICUs, which is not the focus of our study.
Reviewer 3: - row 61 - it is worth mentioning the more rapid methods for the etiological diagnosis (e.g. molecular)
Response: we thank the reviewer for this observation. We added this part (Lines 63-66, highlighted.)
Reviewer 3: - the introduction should include the aim of the study
Response: we thank the reviewer. However, the aim was already written. We only changed “attempted” to “aimed” (Lines 99-103, highlighted).
Reviewer 3: - in the introduction/material and methods, it is worth including the definition of sepsis or MDR/XDR
Response: we thank the reviewer for the suggestions. We included definitions of MDR/XDR infection (Lines 142-145, highlighted). The diagnosis of sepsis is already included (Lines 166-168).
Reviewer 3: - in the material and methods section it should be included further data on the microbiological methods used
Response: we thank the reviewer, new subsection “2.2. Diagnosis of bloodstream infection and microbiological tests” is included in the Material and methods (Lines 154-162, highlighted).
Reviewer 3: - any information about the type of infection (eg community-acquired vds hospital-acquired and the impact on the outcome/etiology?)
Response: we thank the reviewer. We performed additional analyses related to community-acquired and hospital-acquired infections (Table 3, Table 4 and lines 60-62 (Introduction), lines 194, 199 (Methods), lines 271, 315 (Results), and lines 395-406 (Discussion) (all highlighted).
Reviewer 3: - unfortunately, I am not able to view the supplementary files. I suggest adding the tables in the main manuscript, if they are not too big (the manuscript in its current form is not too long and could include further information).
Response: we thank the suggestion. We added Table S1 and Table S2 from the Supplementary Materials into the main text as suggested. Now they are Table 3 and Table 4.
Reviewer 3: - the reference list is scarce and not edited according to the mdpi pattern.
Response: we thank the reviewer. We have added more references on the list and included the explanations in the text (reference list, highlighted). The Instruction for authors says that references may be in any style, provided that author uses the consistent formatting throughout.
Reviewer 3: Overall the manuscript is well-written/documented and could be published after some adjustments.
Response: we thank the reviewer for all his/her constructive suggestions.

Round 2
Reviewer 2 Report
Comments and Suggestions for Authors
Thank you for your reply.
Author Response
Thank you.
Sincerely,
the authors.

Reviewer 3 Report
Comments and Suggestions for Authors
I appreciate the author's efforts in addressing my comments. The quality of the manuscript has significantly improved. I have just a few minor comments:
row 67 - two gram-negative agents -> agents active against gram-negative bacteria
rows 68-69 - anti-gram-negative antimicrobials with dual coverage - specific examples should have been better
row 70: such as.....
rows 133-134 - Vancomycin was added to patients with risk factors for methicillin-resistant S. aureus (MRSA) infections. - such as.........
The definition of CAI/HAI should also be inserted in the material and methods section.
However, another proposed study https://doi.org/10.1016/j.jiph.2022.07.009 analyzes and compares core antimicrobial resistance prevention measures (antibiotic stewardship and infection prevention and control programs) in different ICUs, which is not the focus of our study. - even though this is not the main focus of this current study, at least brief information on this topic would have been a great assessment when discussing infections and AMR.
the reference list is not edited according to the mdpi pattern.
I commend the author for the diligent effort put into this study.
Kind regards,
Author Response
Reviewer 3:
I appreciate the author's efforts in addressing my comments. The quality of the manuscript has significantly improved. I have just a few minor comments:
- row 67 - two gram-negative agents -> agents active against gram-negative bacteria
Response: We thank the reviewer for his/her comment. We have accepted the suggestion and changed (line 72, highlighted).
- rows 68-69 - anti-gram-negative antimicrobials with dual coverage - specific examples should have been better
Response: We accepted the suggestion. We have stated specific examples (lines 73-75, highlighted).
- row 70: such as.....
Response: We accepted the suggestion and added examples (lines 76-77, highlighted).
- rows 133-134 - Vancomycin was added to patients with risk factors for methicillin-resistant S. aureus (MRSA) infections. - such as.........
Response: We accepted the comment and supplemented the sentence (lines 142-144, highlighted).
- The definition of CAI/HAI should also be inserted in the material and methods section.
Response: We inserted the definition of CAI/HAI in the Material and Methods section (lines 156-161, highlighted).
- However, another proposed study https://doi.org/10.1016/j.jiph.2022.07.009 analyzes and compares core antimicrobial resistance prevention measures (antibiotic stewardship and infection prevention and control programs) in different ICUs, which is not the focus of our study. - even though this is not the main focus of this current study, at least brief information on this topic would have been a great assessment when discussing infections and AMR.
Response: We accepted the suggestion and added brief information on this topic (lines 63-67, highlighted).
- the reference list is not edited according to the mdpi pattern.
Response: We accepted the suggestion and edited the reference list according to the mdpi pattern.
I commend the author for the diligent effort put into this study.
Response: we thank the reviewer again for all constructive suggestions.
Sincerely,
the authors.
